# How patients experience nurse-doctor collaborative care at specialist clinics: A qualitative study

Yang Yann Foo[1]*, Xiaohui Xin[2], Qianhui Cheng[1,3], Hwee Kuan Ong[4,5], Ting Ting Yeoh[6], Wentao Zhou[7,8], Siti Rohaida Rahmat[8], Nigel C.K. Tan[9], Jai Rao[1,10], Kirsty J. Freeman[11], Sok Mui Lim[5], Kevin Tan[9]

1 Duke-NUS Medical School, Singapore, 2 Health Services Research Unit, Singapore General Hospital, Singapore, 3 Department of Neuroradiology, National Neuroscience Institute, Singapore, 4 Department of Physiotherapy, Singapore General Hospital, Singapore, 5 Health & Social Sciences, Singapore Institute of Technology, Singapore, 6 Department of Pharmacy, National Cancer Centre Singapore, Singapore, 7 Alice Lee Centre for Nursing Studies, Yong Loo Lin School of Medicine, National University of Singapore, National University Health System, Singapore, 8 Department of Neuroscience Nursing, National Neuroscience Institute, Singapore, 9 Department of Neurology, National Neuroscience Institute, Singapore, 10 Department of Neurosurgery, National Neuroscience Institute, Singapore, 11 The Rural Clinical School, University of Western Australia, Australia

* yangyann.foo@duke-nus.edu.sg

## Abstract

### Introduction

Patient experience of interprofessional collaboration in primary care has been well-studied but not in specialist clinics. Our qualitative study aimed to understand patients' experience of a nurse-doctor collaboration at three specialist clinics (Epilepsy Clinic, Neuroimmunology Clinic, and Persistent Concussion Clinic) in a tertiary neurology care centre in Singapore.

### Methodology

Between December 2023 and April 2024, participants of different demographic and disease profiles from the three specialist clinics were recruited using maximum variation selection. We generated observation and interview data to understand patient experience in a multifaceted and in-depth manner. We analyzed the data using Braun and Clarke's reflexive thematic analysis.

### Results

We observed 27 patients, of whom 12 agreed to be interviewed. We constructed two themes. The first discussed the patients' varied receptivity to interprofessional collaboration depending on their perceived healthcare needs. Most patients valued collaborative care as it saved time and enhanced their access to psychosocial and financial support. However, patients whose disease status was still active preferred

**Data availability statement:** All relevant data are within the paper and Supporting Information files. More information is available from the SingHealth Central Institutional Data Access / Ethics Committee (contact via yangyann.foo@duke-nus.edu.sg) for researchers who meet the criteria for access to confidential data.

**Funding:** The Lee Foundation, Singapore

**Competing interests:** No competing interests to declare.

to consult the doctors for symptomatic management through drug treatment. They were observed to be reticent about sharing their preference with the care team. The second theme examined the absence of formal introduction of the concept of interprofessional collaboration to the patients. Some patients appeared to be unaware that specialist nurses were qualified to collaborate with doctors, and this lowered their perceptions of the nurses' competence and seemingly weakened their receptivity to IPC.

## Conclusion

Patients' experience of IPC at specialist clinics varied depending on patients' perceived healthcare needs. To optimize patients' receptivity to IPC, the provision of collaborative care should be calibrated to fulfill different patients' perceived and actual healthcare needs. Doing so may optimize the value of collaborative care to patients. Further enhancements to patients' receptivity would involve the intentional effort to prepare patients for collaborative practice.

## Introduction

Interprofessional collaboration (IPC) is a model of care whereby healthcare providers of different professions (doctors, nurses, allied health professionals) work with one another, patients and their families to deliver high quality care [1]. IPC has been heralded as an effective way to provide patient-centred care [2], which implies that patients are the ultimate beneficiaries of IPC. Therefore, it is important to study patients' experience of IPC [3].

Patient experience is not only important in its own right from a humanistic perspective [4], it is also important because patient experience is associated with clinical safety and effectiveness [5]. It has been shown that healthcare services and patient health outcomes are enhanced by positive patient experience. Patients with positive experience tend to have better health outcomes as they are more likely to adhere to medical advice, attend follow-up appointments, and manage their health conditions effectively [6]. Since IPC is intended to be patient-centred, it is reasonable to assume that IPC would generate positive patient experience which would in turn lead to favorable patient health outcomes. However, definitive association between IPC and positive patient outcomes have not been demonstrated [7,8]. This raises the question of how IPC in practice is actually experienced by patients. Our study sought to examine patient experience in specialist clinics that adopted a nurse-doctor collaborative care model.

Collaborative care models between nurses and doctors have previously been examined, especially in their care team dynamics, and continue to be widely adopted [9]. One prior study demonstrated how nurse-doctor collaborative partnership enhanced the satisfaction of patients who received education and counselling advice from nurses complementing time-poor doctors who could not offer those

services [10]. More recently, patients were reportedly satisfied with the greater convenience of care as a result of replacing one doctor's visit with a telephone follow-up by a nurse [11,12]. However, amidst these positive findings also existed ambivalent views – while acknowledging the excellent quality of care provided by nurses, some patients still preferred to be seen by doctors [13] especially for medical aspects of care [14].

A recent survey involving more than 870,000 patients in primary care in England [15] showed that ignoring patient preference when implementing IPC undermined patients' confidence and trust in healthcare professionals (HCP). When patients requested to see a doctor but saw a nurse instead or vice versa, their ratings of confidence and trust in the HCP were both lower than when their original requests were successfully fulfilled (7 and 5 points lower respectively). Based on these findings, Paddison and colleagues cautioned that policymakers should consider the potential unintended consequences for patient experience with the widespread introduction of multidisciplinary teams in primary care.

While patient experience of IPC in primary care has been well-studied [16], the nurse-doctor collaborative partnership in a specialist care setting is contextually different. Even though both groups of nurses provide social support and education to patients, primary care nurses provide general care education and specialist nurses provide speciality care education which requires advanced skills. Such contextual differences call for novel research to examine the impact of nurse-doctor collaboration on patient experience in a specialist setting. Our qualitative study aimed to understand patients' experience of a nurse-doctor collaboration at three specialist clinics in a tertiary neurology care centre, hence presenting a novel contribution to existing literature.

## Methodology

### Setting

We studied three specialist clinics at the National Neuroscience Institute, a tertiary centre in Singapore which cared for patients with neurological conditions. The Epilepsy, Neuroimmunology and Persistent Concussion clinics were identified as engaging in IPC practices in an earlier study [17]. The collaboration in these three clinics comprised specialist nurses (Advanced Practice Nurse or Nurse Clinician), and doctors with specialist training (an epileptologist, a neuroimmunologist or a neurosurgeon). These HCP for each of the specialist clinics were members of a fixed team and see patients in the same location and session concurrently. Each of the specialist clinics organized pre- or post-consultation team meetings to discuss patients' assessment and management. This study was part of a larger project that sought to measure IPC's return of investment underpinned by the Quadruple Aim [18].

All three specialist clinics have patients seen by both nurses and doctors. Epilepsy Clinic and Neuroimmunology Clinic look after patients with chronic diseases with acute exacerbations while Persistent Concussion Clinic cares for patients with a more acute condition of mild head injury (MHI) which may have chronic consequences. This arrangement provides holistic care for patients so that they could receive education and counselling by the specialist nurse before being seen by the doctor.

### Recruitment and participants

SingHealth Central Institutional Review Board provided review exemption for this study (CIRB Ref: 2023/2548). The exemption status was granted based on the condition that this study was part of a larger project evaluating the return of investment of the three IPC practices and only anonymous data from the participants were used for this study. Participants were recruited with written informed consent by QC, XX and YYF who had no prior relationship with or influence over the patients at the specialist clinics. This ensured that patients participated voluntarily in the study.

Participants were purposively selected: they must be receiving care at the three specialist clinics with IPC practice. Additional criteria included the ability to speak English or Mandarin, and the ability to give informed consent or have caregivers who could serve as proxy participants. Participants of different demographic and disease profiles were recruited

using maximum variation selection strategies (Table 1). We chose this strategy as it was most likely to generate rich insights into a phenomenon by exploring the different perspectives of participants with diverse characteristics [19].

## Definitions

**Patient experience.** There is little consensus regarding the definition of patient experience. Some scholars [20,21] have deemed the most comprehensive version as that where patient experience was defined as the "sum of all interactions, shaped by an organizations' culture, that influences patients' perceptions across the continuum of care" [22,23]. We found this definition too broad and challenging to operationalize. Instead, we chose a narrower definition which examined both "patients' experiences of care" as well as the "feedback received from patients about those experiences" [24]. Based on the definition offered by Ahmed and colleagues [24], we sought to understand patient experience through interviews, to gather patients' *accounts* of their care experience (e.g., "What did the nurse/doctor talk to you about?") and their *evaluation* of that experience (e.g., "How did you feel about receiving care advice from the nurse/

**Table 1. Characteristics of participants.**

|  | Observed | Interview status | Pseudonym | Age | Gender | Disease status[+] | Clinic[*] |
|---|---|---|---|---|---|---|---|
| 1. | Yes | Yes | Masud | 30s | M | Active | A |
| 2. | Yes | Declined | NA | 40s | M | Active | A |
| 3. | Yes | Recruited but no show | NA | 60s | F | Inactive | A |
| 4. | Yes | Yes | Anushka | 20s | F | Inactive | A |
| 5. | Yes | Yes | Charlotte | 50s | F | Active | A |
| 6. | Yes | Yes | Linda | 80s | F | Inactive | B |
| 7. | Yes | Recruited but no show | NA | 30s | F | Inactive | B |
| 8. | Yes | Yes | Nancy | 60s | F | Inactive | B |
| 9. | Yes | Yes | Siti | 40s | F | Inactive | B |
| 10. | Yes | Yes | Yusuf | 20s | M | Inactive | B |
| 11. | Yes | Yes | Mabel | 40s | F | Inactive | B |
| 12. | Yes | Declined | NA | 70s | F | Inactive | C |
| 13. | Yes | Declined | NA | 40s | M | Inactive | C |
| 14. | Yes | Declined | NA | 40s | M | Inactive | C |
| 15. | Yes | Declined | NA | 20s | M | Active | C |
| 16. | Yes | Declined | NA | 40s | M | Inactive | C |
| 17. | Yes | Declined | NA | 50s | F | Active | C |
| 18. | Yes | Declined | NA | 60s | F | Inactive | C |
| 19. | Yes | Declined | NA | 70s | F | Inactive | C |
| 20. | Yes | Declined | NA | 30s | M | Inactive | C |
| 21. | Yes | Declined | NA | 20s | F | Inactive | C |
| 22. | Yes | Yes | Gerald | 60s | M | Active | C |
| 23. | Yes | Yes (parent of cognitively challenged patient in her 40s) | Tony | 70s | M | Inactive | C |
| 24. | Yes | Yes | Sabrina | 40s | F | Active | C |
| 25. | Yes | Declined | NA | 20s | F | Inactive | C |
| 26. | Yes | Recruited but no show | NA | 60s | F | Inactive | C |
| 27. | Yes | Yes | Rose | 70s | F | Inactive | C |

[+]Active refers to the state of a disease with intermittent flare-ups or exacerbations; inactive refers to the state of a disease with no flare-ups or exacerbations within the past 12 months.

[*]For the purpose of anonymity, the actual clinic is not identified.

doctor?"). Criteria used to differentiate between positive and negative patient experience included dimensions such as respect, information and communication, physical comfort, emotional support, and access to care [24].

**Collaboration.** The practice of IPC is far from being homogeneous [25,26] and may range from loose networking to highly integrated teamwork [27]. Conceivably, patient experience of different models of IPC may vary. To identify the kind of IPC practiced at the specialist clinics that was experienced by our participants, we asked HCP from the three specialist clinics to independently describe their team's IPC type using the InterPACT framework [9]. The HCP collectively reflected on InterPACT's six dimensions (shared commitment, shared team identity, clear goals, clear team roles and responsibilities, interdependence between team members, and integration between work practices) and noted the extent to which each dimension characterized their way of working. They chose "collaboration" which matched the independent assessment of XX and YYF who collected observational data at the three specialist clinics.

## Data generation

Between December 2023 and April 2024, XX and YYF collected observation data, and QC, XX, and YYF collected semi-structured interview data. The duration of observations was three hours on average, and interviews ranged between 30 and 60 minutes, averaging 5500 words. To minimize disruptions, marginal participant approach [28] was used to observe patient interactions and communication with HCP. To remain unobtrusive, XX and YYF positioned themselves at the furthest distance where they could still clearly hear the conversations between patients and HCP. Our observation tool (S1 Appendix) and interview guide (S2 Appendix) were developed using Ahmed and colleagues' [24] definition of patient experience and existing literature reviewed in the introduction. Revisions were made to the interview guide as we collected and analyzed data iteratively. Observational field notes were recorded by hand, and semi-structured interviews were audio-recorded with consent and transcribed verbatim manually by QC. This dual approach of data generation enabled crystallization, a process of gathering multiple types of data to explore the phenomenon under study in a more multifaceted and in-depth manner [29].

The decision to stop data generation was based on the recommendations made by Braun and Clarke [30], aligned with our contextualist ontology [31] whereby our research philosophy was situated, interpretivist and reflexive. We accept that different meanings are constructed by different researchers depending on their situated subjective experiences [32,33]. Therefore, we do not subscribe to the (post)-positivist concept of data saturation where data collection ceases only when no new data are generated [34]. Instead, we assessed the depth of data generated from each participant and the perspectival diversity in that data [30]. After 27 observations and 12 interviews, we determined that we have generated adequate data to provide a rich, complex and multi-faceted account [35] of how patients experienced nurse-doctor collaborative care at the three specialist clinics.

## Data analysis

We analyzed the data using Braun and Clarke's reflexive thematic analysis, as it is congruent with our ontology of contextualism [31]. This meant that our interpretation of the participants' experience was mediated through the lens of sociocultural realities [31]. We chose reflexive thematic analysis as it is a suitable method for our research question which sought to understand participants' experiences [31].

We followed the multi-phase analytic approach described by Braun, Clarke and colleagues [31]. All or some of the transcripts were read and re-read by QC, HKO, XX, YYF, and YTT, and each of us generated separate familiarization notes. After discussing our initial impressions, we engaged in semantic (descriptive) and latent (interpretative) coding.

We then clustered these codes around central organizing concepts to construct candidate themes. We iteratively reviewed and finalized the themes in a consultative manner over a total of ten group meetings. At these meetings, we discussed how we made sense of the data based on our backgrounds, experience and knowledge. Whenever when we had differing perspectives regarding the themes, we consistently revisited our study's research aim and

also scrutinized the data. The perspective that prevailed would be the one which most compellingly advanced our understanding of patients' experiences with a nurse-doctor collaboration at specialist outpatient clinics and was strongly supported by data. This approach enabled us to determine how well these themes provided an account of all our data.

We initially constructed three themes. The first was "Patients' awareness of and receptivity to collaborative care", the second "Cognitive dissonance due to perceived mismatch to patients' healthcare needs", and the third "Absence of formal introduction of IPC to patients". However, during the review process, we found overlaps between the first two themes: both were about patients' reception to IPC, just that the first theme described positive reception and the second theme mixed reception. From the standpoint of central organizing concepts, this was a weakness. To disambiguate this overlap, we defined both themes and concluded that it would be more coherent if we revised theme 2 to be a sub-theme of theme 1. Using this process, we finalized the analysis and constructed the two final themes (see Analysis).

In the last phase where we produced the report, we replaced participants' names with pseudonyms to protect their confidentiality. To provide context in a succinct manner, we tagged the data extracts with our participants' gendered names, age range and the current state of their disease status.

Our analysis adhered to Braun and Clarke's thematic analysis reporting guidelines [36] (S3 Appendix) congruent with our research values [33].

**Reflexivity.** Congruent with our contextualist ontology and for the purpose of adhering to the quality criterion of trustworthiness, we will highlight relevant aspects of our sociocultural realities (i.e., background, prior experiences with IPC etc.) that may have influenced our interpretation of the data. HCP members in the research team provided inputs from the perspectives of nursing (RR, ZW), allied health (OHK, SML, YTT), and medicine (JR, KT, NCKT). Non-HCP members' interpretations of the data were based on their positionalities as health services (QC, XX) and qualitative (YYF) researchers, and also as patients who have experienced IPC. For reasons of confidentiality, JR, KT and NCKT did not read the transcripts as the participants were attending their practice. They provided inputs to the manuscript draft, after the names of the participants had been replaced with pseudonyms.

## Results

We observed 27 patients from Epilepsy Clinic, Neuroimmunology Clinic and Persistent Concussion Clinic, of whom 12 agreed to be interviewed (Table 1). Using the criteria of the depth of data generated from each participant and the perspectival diversity in that data, we determined that we have generated adequate data to understand how patients experienced nurse-doctor collaborative care at the three specialist clinics.

We constructed two themes related to how patients experienced the nurse-doctor collaboration. The first theme examined patients' variable receptivity to IPC depending on their perceived healthcare needs. Theme 1 also has a sub-theme, which described patients' reticence to share their preference with either the nurse or the doctor. Theme 2 examined the absence of formal introduction of IPC to patients, and the probable impact of this on some patients' perception of the competence of and role played by specialist nurses.

### Theme 1: Receptivity to IPC varied depending on patients' perceived healthcare needs

In this theme, we describe how the participants expressed their receptivity to the fixed, co-located nurse-doctor collaboration model of IPC. While most participants in general commented positively about the specialist doctors' expertise and professionalism, their responses to the specialist nurses were more varied. Many spoke appreciatively about how specialist nurses improved access to holistic care. However, some expressed mixed feelings about the delivery of nurse-doctor collaborative care.

Participants with positive IPC experiences spoke about how well the nurses complemented the specialist doctors to provide them holistic care. To these participants, this was care that transcended the medical treatment of the disease.

One of its multiple dimensions was the psychosocial support provided by the specialist nurses to assuage the participants' anxiety about how to live with their illnesses:

*Back then, it was still during Covid but they were transiting back to in-person (activities), and I had some concerns about whether that level of exposure was good for me in terms of infection risk. (I feel that for such) personal concerns, nurses might be able to understand more of (than doctors) what the patient is experiencing. Like whether it's valid for me to have this level of concern.* (Anushka; 20s; disease status: inactive)

Some participants also highlighted several other aspects of holistic care. These included being treated as a person, gaining greater access to care continuity, and benefiting from the specialist nurses' comprehensive and useful knowledge about how to access financial support for patients:

*Every time the doctor needs to recall what's my case, what's my history, what are my scans but the nurse knows everything about me – she remembers my name. It's easier to see the nurse: I can contact her. There was one time I was running out of medicine. So I cannot see him (the doctor), I will need an appointment. So it's easy to contact her to (refill my medication). The nurse is more familiar with who's going to help, who is responsible for this, who's responsible for that. For example, recently I started to have some financial support for the medicine. Dr (so-and-so) knows that there's financial support, but he doesn't know what to be done. So the nurse is the one who can do this.* (Masud; 30s; disease status: active)

**1.1 Patients' reticence to share their reservations about IPC.** Even though IPC was appreciated by participants who needed and benefitted from nursing collaboration, those who did not have such perceived needs expressed varying degrees of reservations about the model of nurse-doctor collaborative care. Interestingly, while they would share their reservations with the interviewers, during their consultation, they were observed to be reticent about sharing such feelings to the nurse or doctor.

Among participants who felt that holistic care was less relevant to them, some wondered if the nurse's role was to "help the doctor' manage the patient load". (Gerald; 60s; disease status: active). Others also felt that there was an overprovision of care by way of redundancies:

*(The doctor said) three sentences, the same sentence as the nurse gave, so it's the same you see. Ya, wasting time this and that.* (Linda; 80s; disease status: inactive)

There was another group of participants who expressed mixed feelings about the way IPC was delivered. These participants' disease status was still active, and they shared that they preferred to consult the doctors for symptomatic management through drug treatment. While this group of patients acknowledged that the nurses were experienced and could help to save precious waiting time, they still felt uneasy about receiving medical inputs from nurses:

*(The nurse) can do the summary of, like the notes for my condition. It can be a win-win situation. It can be good for Dr (so-and-so), it can also be good for me. It could be like time saving for both ways. But to (have the nurse) talk to me about increasing the dosage of my medication? I was quite apprehensive. Honestly, considering my other conditions right, I have to (take multiple) pills a day. I know she's definitely experienced to be in her role. But maybe I should speak to my doctor first (laughs). I would prefer Dr (so-and-so) to diagnose my problem and come to me with (a) solution (related to medication and treatment issues).* (Sabrina; 40s; disease status: active; has morbidity)

As can be seen by the above data extract, during the interview, Sabrina clearly articulated her reservations about receiving medication dose adjustment advice from the nurse. However, she did not express these feelings to the nurse. Instead,

based on our observational data and memos, Sabrina's coping mechanism seemed to be to "repeat, four times, her concern that being comorbid, she was worried about taking a higher dose of medication, especially since earlier increments had not eased her disease symptoms" (Field notes).

Sabrina appeared to be reticent when interacting with the doctor. During the interview, Sabrina emphasized how it was important to her that the doctor made medical "decisions" and not be "influenced" by the nurse's recommendations. She spoke about her guilt of burdening family members because her disease was still active, and how she relied on the doctor to get better through medical means. But we did not observe any attempt on her part to communicate this to the doctor. The only indications Sabrina let on about how she felt were "her slumped shoulders and subdued speaking voice towards the end of the consultation with the doctor" (Field notes). Sabrina's reticence was not unique – a similar pattern was observed among other participants such as Gerald and Linda.

### Theme 2: Absence of formal introduction of IPC to patients

As IPC was a less familiar concept to patients than uniprofessional care where they were seen by HCP individually, we sought to understand how the participants understood the rationale and purpose of the nurse-doctor collaborative care they were receiving. They reportedly "cannot recall" being formally introduced to IPC (Charlotte; 50s; disease status: active). Interestingly, however, even without formal introduction, a few participants seemed to have closely observed the interactions among the HCP and thus were aware that the care provided at the three specialist clinics was more collaborative in nature:

*Based on my experience with your healthcare professionals (nurse and doctor), I think they read the notes that are written (by each other), they trust each other. They actually (called each other) by the first name so I believe they know each other very well. I believe it's probably a close community. I do feel that they are actually one entity.* (Yusuf; 20s; disease status: inactive)

The collegial and collaborative interactions between the respective nurse-doctor teams were also observed and recorded in the field notes:

*It's interesting how all three doctors from the specialist clinics seem to stand whenever they go to the nurse's room to see patients. Is this deliberate? Is this done so that there's minimal disruption to the conversation the nurses have been having with patients? Also, do the doctors consciously want to come across as not being more senior in hierarchy to the nurses who remain seated while they stand? (Field notes)*

However, perhaps because the rationale and purpose of IPC had not been formally introduced, the un-initiated patients who saw the nurse first appeared to have experienced mixed emotions, even if their encounter with the nurse had been positive:

*I was thinking, I came all the way, to see a doctor, but yet I met the nurse. Although the nurse also did a fantastic job, very assuring, and very clear in what she wanted to tell me, but it will be good that she tell me, Dr (so-and-so) is now busy with (another)patient, I will start this session first. At least put things in proper perspective that actually there is a doctor, apart from her.* (Nancy; 60s; disease status: inactive)

Another disadvantage of the absence of formal introduction to IPC was the missed opportunity to educate patients about the changing and expanded role played by specialist nurses. While some patients have intuitively figured out that Advanced Practice Nurses and Nurse Clinicians were "more specialized" and possessed "greater medical knowledge of specific conditions" (Anushka; 20s; disease status: inactive), many still thought of nurses as "sweet and nice" (Rose;

70s; disease status: inactive) who were "doctors' assistants" (Tony; 70s; parent of patient; disease status: inactive). In the minds of these patients, nurses were HCP who did administrative tasks such as "record immediately any changes to the medicine or (remind the doctor) to give the patient 'mc' (medical leave) this kinda thing (because) it's more of an admin job (and) doctor don't do this kinda thing" (Charlotte; 50s; disease status: active).

In sum, in the absence of formal introduction of IPC to patients, most of them did not realize that specialist nurses were trained and qualified to collaborate with the doctor to offer enhanced care for them at the specialist clinics. This lack of awareness appeared to underpin some of the participants' lower expectations of nurses' competence, which could weaken the impact of IPC.

## Discussion

Our qualitative study sought to examine patients' experience of a nurse-doctor collaborative care model in the specialist clinics. We constructed two themes where the first showed that receptivity to nurse-doctor collaborative care was variable. A sub-theme of Theme 1 was how some patients were observed to be reticent about expressing their reservations about IPC to their healthcare professionals. In theme 2, we explored how in the absence of formal introduction to IPC, some patients may have a narrower view of the nurses' scope of practice as they seemed unaware of the nurses' specialist training.

This paper adds to the literature by suggesting that while IPC undoubtedly enhances holistic care, its delivery should be more finely calibrated to fulfill different patients' perceived and actual healthcare needs. This may enhance patients' receptivity to IPC and optimize its value to patients.

For patients whose disease has remained active and who are more focused on managing their disease symptoms, the specialist nurses' role in medication counseling may need to take place after the patient has seen the doctor. This contrasts with the typical mode of nurse-doctor collaboration in specialist clinics where IPC is delivered through having the specialist nurse complete the majority of the initial information gathering, medication reconciliation and counselling so that the doctor would have more time to "consider subtleties of clinical practice as well as time to counsel patients" [37]. Based on our data, this model of care delivery posited by Hill and colleagues should be modified to match patients' perceived and actual needs. Clear explanations will need to be given to patients as they may not be aware that specialist nurses are trained to prescribe medication and adjust doses [38,39]. Without such knowledge and understanding, patients may perceive the nursing collaboration to limit their interactions with doctors [13] who are deemed to possess higher levels of knowledge and judgement [40,41].

In contrast, patients who needed psychosocial support [42] and greater access to care [43] were highly appreciative of specialist nurses' availability to them, leading to the development of therapeutic nurse-patient relationships [44]. Having specialist nurses involved in IPC enabled the integration of their advanced clinical skills and specialized knowledge, complementing physician's consultations and enhanced the patients' overall experience with the specialist care provided. According to our data, these patients tended to be younger and seemed more aware of the training specialist nurses have undergone. Patients aged 50 and above were comparatively less aware of specialist nurses' advanced training. They appeared more fixated on discussing their medical problems and changes in treatments with doctors [45]. This may be because, for most of their lives, they have experienced a different model of care which was uniprofessional and doctor-centric. As such, they may need time to understand and embrace a new, collaborative nurse-doctor model of care.

While delivering bespoke IPC care to patients may enhance their receptivity, it may be challenging to ascertain their perceived needs. Among patients who feel comfortable explicitly expressing their needs [46], this would be easier. However, some patients, such as those of Asian ethnicity, may be more reticent about verbalizing their needs presumably because they have a deferential attitude towards healthcare professionals [47–51]. Hence, it is important to cultivate a certain level of self-advocacy among these patients [21] so that they could become active partners in their healthcare [52]. One model that might be useful in facilitating change in reticent patients is the 4P-tool called Patient Preferences

for Patient Participation [53]. The 4P-tool is a guide comprising 12 steps and could be used to facilitate conversations between HCP and patients around issues such as having conditions for reciprocal communication.

Our findings also underscore the importance of the intentional development of patient-readiness for collaborative practice. Preparing patients to participate in IPC is similar to helping health professions students and HCP to be collaboration-ready through interprofessional education programs [54,55] and interprofessional competency frameworks [56–59]. Getting patients ready for IPC through helping them understand how collaborative practice might enhance their care is arguably essential since most may not be aware that specialist nurses, such as the APN, receive rigorous training and go through stringent accreditation process to qualify to engage in an expanded scope of practice [60]. Our study suggests that patients who are 50 and older seem less aware that nursing training has made significant advancements and thus appeared to assign less value to IPC. Thus, the focus of patient-readiness initiatives could be on engaging patients who are 50 and older first before extending to other groups.

A key component of such patient-readiness initiatives could include training the specialist nurses to clearly communicate and explain the role they play in patient care [61] and also for doctors to explain IPC to patients. Another, perhaps even more significant, component will be patients' formal introduction to IPC by the nurse-doctor team. They could jointly meet the patients and explain their complementary roles meant for the enhancement of care. Also helpful will be mass media publicity efforts to improve the professional image of nursing and increase the public's understanding of nursing [38]. Other initiatives could also include the development of structured materials, orientation techniques and patient feedback mechanisms that would enhance patient engagement and understanding of the nurse-doctor collaborative care model.

Establishing an IPC clinic requires a clear, justifiable rationale for its purpose, emphasizing the value that speciality nurses bring to patient experiences, beyond simply reducing doctors' workload. Defining the specific roles of specialist nurses within an IPC clinic and ensuring that they are properly introduced by nurses and doctors themselves can help prevent patient misperceptions of overlapping practices, reduce role ambiguity, and promote patients' receptivity to IPC.

In the context of the nurse-doctor collaborative care model, patient awareness of such developments is especially crucial. In Australia, when patient-participants became aware of the scope of practice of Nurse Practitioners (highly trained nurses who are board-certified), nearly 92% agreed to be seen by them [62]. Just as role clarification has played a crucial part in fostering collaboration among different health professions [63,64], clarifying the nurses' role in IPC would also help patients be more ready to appreciate the nursing contributions. As has been shown, patients cognizant of specialist nurses' capabilities are highly receptive to their care [65].

Moving forward, it is essential to formally introduce the concept of IPC to patients at the outset of their treatment journey to promote transparency and establish trust in IPC care. This may be especially important for specialty clinics as patients may expect to receive care from only specialist doctors. Developing communication strategies that engage patients in understanding the role of the IPC clinic should be considered. This approach can empower patients, enhance their experience, and encourage their involvement in their own care, and most importantly constantly improve the IPC care. Afterall, patients are an important part of the healthcare team [1,66].

## Limitations of the study

Numerous specialist clinics caring for other medical conditions (such as stroke and dementia) exist at the national centre and our study examined only three of them. To deepen our understanding of patient experience of specialist care, more studies should be conducted at the other clinics.

## Conclusion

Patients' experience of IPC at specialist clinics varied depending on patients' perceived healthcare needs. To optimize patients' receptivity to IPC, the provision of collaborative care should be calibrated to fulfill different patients' perceived

and actual healthcare needs. Doing so may optimize the value of collaborative care to patients. Further enhancing of IPC receptivity would involve the intentional development of patient-readiness initiatives for collaborative practice. This study offers a novel contribution to extending the current understanding of IPC in specialist contexts.

## Supporting information

**S1 Appendix. Observation tool.**
(DOCX)

**S2 Appendix. Interview guide.**
(DOCX)

**S3 Appendix. Reporting guidelines.**
(DOCX)

## Acknowledgments

The research team would like to acknowledge The Lee Foundation for funding this study. We would also like to express our gratitude to all the participants who took part in this study for sharing their experiences.

## Author contributions

**Conceptualization:** Yang Yann Foo, Xiaohui Xin, Qianhui Cheng, Hwee Kuan Ong, Nigel C.K. Tan, Jai Rao, Kirsty J. Freeman, Sok Mui Lim, Kevin Tan.

**Data curation:** Qianhui Cheng.

**Formal analysis:** Yang Yann Foo, Xiaohui Xin, Qianhui Cheng, Hwee Kuan Ong, Ting Ting Yeoh.

**Funding acquisition:** Nigel C.K. Tan.

**Investigation:** Yang Yann Foo, Xiaohui Xin, Qianhui Cheng.

**Methodology:** Yang Yann Foo.

**Project administration:** Kevin Tan.

**Writing – original draft:** Yang Yann Foo.

**Writing – review & editing:** Yang Yann Foo, Xiaohui Xin, Qianhui Cheng, Hwee Kuan Ong, Ting Ting Yeoh, Wentao Zhou, Siti Rohaida Rahmat, Nigel C.K. Tan, Jai Rao, Kirsty J. Freeman, Sok Mui Lim, Kevin Tan.

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
