## [Decision Letter · Decision Letter 0]

26 Jan 2025

PONE-D-24-54289How patients experience nurse-doctor collaborative care at specialist clinics: A qualitative studyPLOS ONE

Dear Dr. Foo,

Thank you for submitting your manuscript to PLOS ONE. After careful consideration, we feel that it has merit but does not fully meet PLOS ONE’s publication criteria as it currently stands. Therefore, we invite you to submit a revised version of the manuscript that addresses the points raised during the review process.

**ACADEMIC EDITOR: **

We look forward to receiving your revised manuscript.

Kind regards,

Mohd Ismail Ibrahim, MCom.Med

Academic Editor

PLOS ONE

2. Thank you for stating the following financial disclosure:  [The Lee Foundation, Singapore]. At this time, please address the following queries:

3. In this instance it seems there may be acceptable restrictions in place that prevent the public sharing of your minimal data. However, in line with our goal of ensuring long-term data availability to all interested researchers, PLOS’ Data Policy states that authors cannot be the sole named individuals responsible for ensuring data access (http://journals.plos.org/plosone/s/data-availability#loc-acceptable-data-sharing-methods).

Additional Editor Comments (if provided):

Reviewers' comments:

Reviewer's Responses to Questions

**Comments to the Author**

1. Is the manuscript technically sound, and do the data support the conclusions?

Reviewer #1: Yes

Reviewer #2: Yes

Reviewer #3: Yes

2. Has the statistical analysis been performed appropriately and rigorously? 

Reviewer #1: Yes

Reviewer #2: N/A

Reviewer #3: N/A

3. Have the authors made all data underlying the findings in their manuscript fully available?

Reviewer #1: Yes

Reviewer #2: Yes

Reviewer #3: Yes

4. Is the manuscript presented in an intelligible fashion and written in standard English?

Reviewer #1: Yes

Reviewer #2: Yes

Reviewer #3: Yes

5. Review Comments to the Author

Reviewer #1: I appreciate your work and enjoyed reviewing it. I have a couple of comments on the methodology section:

1. Please provide any specific reference on how you decided that a sample of 27 with a response of 12 is enough for generating valid results of your qualitative research work.

2. How was an agreement reached to establish consensus on the themes among the authors?

Reviewer #2: Peer Review:

Study Overview, Abstract and Introduction:

The study under review explores how patients experience nurse-doctor collaborative care in specialist clinics, addressing a significant and evolving care model relevant to both primary and specialist medical communities. It qualifies as original research, offering meaningful contributions to the literature on interprofessional collaboration (IPC). The study's strengths include a robust background context, thoughtful consideration of reflexivity, and efforts to reduce extraneous subjectivity in its methodology.

The abstract effectively outlines the study’s major themes, highlights its focus on thematic analysis, and summarizes the main findings, all while adhering to word count requirements. However, refinement in phrasing is recommended. For instance, while the abstract identifies major points of the study, the statement “major themes within article and guidelines for thematic analysis” could be more explicitly tied to specific findings to improve clarity and impact.

Strengths of the Introduction include:

Clear Structure: The introduction provides a well-organized rationale for the study, establishing the relevance and importance of IPC care models in both literature and clinical practice.

Contextual Relevance: The section integrates prior research on patient experience (PX), trust in healthcare systems, and opposing perspectives on the IPC care model, offering a balanced context for the study.

Research Aim: The research aim is clearly articulated, providing readers with a direct understanding of the study’s purpose and objectives.

Balanced Perspectives: The introduction acknowledges opposing viewpoints regarding PX in IPC care, adding depth and rigor to the discussion.

Critique of the Introduction include:

Ambiguity in Sentence Structure: The sentence, “While some patients appreciated the holistic support offered by the nurse-doctor collaborative care model, patients whose disease status was still active seemed eager to control their symptoms through drug treatment managed by the doctor,” lacks clarity. It implies a dichotomy in patient experiences but does not clearly connect these sentiments to broader themes or findings. Contextualizing in discussion or rephrasing is recommended to ensure the point is well-articulated and supports the study’s framework.

Conclusion Statement: The concluding assertion, “Interprofessional collaboration enhances patient care in specialist clinics,” reads more as an opinion rather than a conclusion supported by evidence from the analysis. Consider revising this statement to objectively reflect specific findings or themes emerging from the study.

Specific Recommendations:

-Clarify the sentence about patient perspectives to ensure it aligns with the study's analytical framework and contributes to the broader discussion of PX in IPC care.

1. Original Research

This study demonstrates originality by investigating patient experiences within the context of interprofessional collaborative nurse-doctor care models in specialist clinics. This is a topic that provides valuable insights into this evolving area of healthcare delivery.

2. Novelty of Results

While there are opportunities to reference the study as amore novel contribution to literature, the presented work builds upon existing research on interprofessional collaboration in primary care settings. However, the study's focus on specialist contexts is less extensively explored in the literature. This suggests a novel contribution to the field by extending current understanding to a relatively understudied area

Specific recommendations:

More explicitly state the novelty of the research in areas of application for reader to understand or in its role of solely being explored in specialty settings.

3. Technical Standards and Methodology

The study appropriately utilized Braun and Clarke's reflexive thematic analysis, demonstrating a clear understanding of reflexivity within the methodological framework. Coding and theme development were well-described. While the rationale for the shift from three initial themes to two final themes with sub-themes was partially explained, further clarification on this decision-making process is recommended. Acknowledging the researchers' own reflexivity provides valuable contextual information that may have influenced data interpretation.

Specific recommendations for methodology that require further attention:

The inclusion of specialist clinics engaging in interprofessional collaboration (IPC) practices is valuable, however, recommend a more detailed description/rationale of the criteria used for specialty clinic selection.

The statement regarding SingHealth Central Institutional Review Board exemption should be clarified to include what specific data categorization applies to this.

The definition of "inactive" in the context of disease state should be further defined for period disease inactivity (e.g., no flare-ups within the past 12 months).

The study should provide more specific details on how the InterPACT framework was applied to categorize IPC within the context of the study's focus and its usage by healthcare professionals (HCPs) in self-assessment. The statement regarding the researchers' efforts to be unobtrusive in patient interactions should be further elaborated upon to clarify the extent of their observation during patient encounters.

Consistent use of grammar tense: Recommend consistent use of past tense throughout. The statement regarding the overlap between the first two themes requires rephrasing for improved clarity. In acknowledging the potential influence of researcher positionality on data interpretation, the use of "may have" instead of definitive statements would be more appropriate (e.g., "Non-HCP members' interpretations of the data may have been based on their positionalities…")."

4. Conclusions and Data Support

The study effectively delineated themes for reporting results. Conclusion was concise with mention of pertinent positives from analysis of data.

The following are specific recommendations:

-Study “strengths” section: Currently there is not a section referencing study strengths. The article could be strengthened by a more explicit discussion of study strengths. The acknowledgment of reflexivity in the research process is a notable strength.

-The choice of methodology, including the maximum variation selection strategy, requires a more explicit discussion of its strengths and limitations.

-Recommend the article acknowledge the potential limitations of reflexivity in the research process.

-Recommend the use of the term "research story" to describe data analysis be replaced with more scientifically rigorous terminology.

There is not a clear definition of "Patient Experience" and the criteria used to differentiate between "positive" and "negative" patient experiences.

-Caution and recommendation to restate use of descriptive terminology, such as "eager" or "reticent," to infer patient motivations to be carefully considered and contextualized to avoid potential misinterpretations.

-Lines 408 - 414 - Re-state terms like "the patient," "these patients," and "this group of patients," making it difficult to understand which specific demographic is being referred to. More consistent and precise descriptions of patient demographics are crucial to avoid misinterpretations.

-Line 422 - The use of the term "older patients" without defining the age threshold should be avoided.

-Line 431 - The statement regarding potential reticence among patients of Asian ethnicity should be further contextualized with specific references to relevant cultural norms within the medical community, if this is applicable to meaning.

-Line 443 - the phrase "Getting patients ready..." should be rephrased or contextualized further, as the definition of "patient preparedness" is not clearly defined in the article.

5. Overall Presentation

Overall, the article is presented in an intelligible manner with a logical sequence and structure that makes it easy to follow. Aforementioned recommendations to avoid subjectivity and for appropriateness in scientific presentation would be advised

6. Ethical Standards and Integrity

The research adheres to ethical standards, as demonstrated by the study design and intervention protocols. Participant consent was appropriately documented, and the use of pseudonyms to protect confidentiality was well explained.

-Further clarification is needed regarding the statement: “SingHealth Central Institutional Review Board provided review exemption for this study (CIRB Ref: 2023/2548).” Specifically, the authors should clarify whether this exemption impacted participant consent or any other ethical considerations related to the study.

7. Reporting Guidelines and Data Availability

Appendix included guidelines and interview tools. There was adherence to Braun and Clarke’s Reflexive Thematic Analysis Reporting Guidelines (RTARG).

Final Assessment

This is a novel study that has relevance in primary care and specialist medical communities. With revisions, as recommended, would support publication within PLOS.

Reviewer #3: Re: How patients experience nurse-doctor collaborative care at specialist clinics: A

qualitative study

Thank you for having me to review this paper.

Here’s some comments on the paper:

- Start line 65: I don’t see the necessary to abbreviate the “patient experience” into PX.

- Lines 73-74: “However, definitive association between IPC and positive patient outcomes have not been demonstrated in systematic reviews”: I think this sentence is only proper when this current study then conducted a systematic review rather than a qualitative study. Consider to paraphrase or remove “…in systematic review”

- Please move Table 1 to be cited somewhere in the results (analysis) section instead in method

- I did not see the reporting guidelines of this paper. Consider to look at Equator Network https://www.equator-network.org/) or COREQ (mentioned in the Equator as well). In COREQ, for example, the authors need to explain the credentials of the researchers and their experience/training in qualitative research methods. It is including who conducted the interviews and data analysis (not only their initials, but the background and their experience in conducting a qualitative studies). Even COREQ mention about the gender of the interviewer and data analysts.

- Line 172: “….averaging 5500 words” please consider to move some parts of the method section into the analysis (as commonly known as results). I found some duplication (such as the mention of Table 1” both in the method and analysis. Therefore, the authors need to consider which parts are methods, and which parts more suitable for the results section.

- Line 179: “…transcribed verbatim by QC” is there any supporting device for the transcribing process?

- Lines 190-192: Also consider to move this part to analysis (results)

- The process of the data analysis has been described clearly and in detail.

- How about the trustworthiness of the data? It should be described somewhere in the method section.

- Start from line 272: Please delete all of the name of the participants for the confidentiality, and consider to use a particular code, i.e. P1 for Participant 1.

- I have no further comment for the discussion part.

Thank you

6. PLOS authors have the option to publish the peer review history of their article (what does this mean? ). If published, this will include your full peer review and any attached files.

**Do you want your identity to be public for this peer review?** For information about this choice, including consent withdrawal, please see our Privacy Policy .

Reviewer #1: No

Reviewer #2: **Yes: ** Nailah Abdulbaaqee MD, MPH

Reviewer #3: No

---

## [Author Response · Author response to Decision Letter 1]

16 Feb 2025

Please see attached file ('Response to Reviewers)

---

## [Decision Letter · Decision Letter 1]

4 Mar 2025

How patients experience nurse-doctor collaborative care at specialist clinics: A qualitative study

PONE-D-24-54289R1

Dear Dr. Foo,

We’re pleased to inform you that your manuscript has been judged scientifically suitable for publication and will be formally accepted for publication once it meets all outstanding technical requirements.

Kind regards,

Mohd Ismail Ibrahim, MCom.Med

Academic Editor

PLOS ONE

Additional Editor Comments (optional):

Reviewers' comments:

Reviewer's Responses to Questions

**Comments to the Author**

1. If the authors have adequately addressed your comments raised in a previous round of review and you feel that this manuscript is now acceptable for publication, you may indicate that here to bypass the “Comments to the Author” section, enter your conflict of interest statement in the “Confidential to Editor” section, and submit your "Accept" recommendation.

Reviewer #1: All comments have been addressed

Reviewer #2: All comments have been addressed

2. Is the manuscript technically sound, and do the data support the conclusions?

Reviewer #1: Yes

Reviewer #2: Yes

3. Has the statistical analysis been performed appropriately and rigorously? 

Reviewer #1: Yes

Reviewer #2: Yes

4. Have the authors made all data underlying the findings in their manuscript fully available?

Reviewer #1: Yes

Reviewer #2: No

5. Is the manuscript presented in an intelligible fashion and written in standard English?

Reviewer #1: Yes

Reviewer #2: Yes

6. Review Comments to the Author

Reviewer #1: .

Reviewer #2: All recommendations have been updated and are satisfactory. While not all data has been made public, a specified explanation for restricted data access is provided in accordance with the PLOS Data Policy. I support the acceptance of this article for publication.

7. PLOS authors have the option to publish the peer review history of their article (what does this mean? ). If published, this will include your full peer review and any attached files.

**Do you want your identity to be public for this peer review?** For information about this choice, including consent withdrawal, please see our Privacy Policy .

Reviewer #1: No

Reviewer #2: **Yes: ** Nailah Abdulbaaqee MD, MPH

---

## [Editor Report · Acceptance letter]

PONE-D-24-54289R1

PLOS ONE

Dear Dr. Foo,

I'm pleased to inform you that your manuscript has been deemed suitable for publication in PLOS ONE. Congratulations! Your manuscript is now being handed over to our production team.

Kind regards,

on behalf of

Dr. Mohd Ismail Ibrahim

Academic Editor

PLOS ONE